# Monitoring the Redox Status in Multiple Sclerosis

**DOI:** 10.3390/biomedicines8100406

**Published:** 2020-10-12

**Authors:** Masaru Tanaka, László Vécsei

**Affiliations:** 1MTA-SZTE, Neuroscience Research Group, Semmelweis u. 6, H-6725 Szeged, Hungary; tanaka.masaru.1@med.u-szeged.hu; 2Department of Neurology, Interdisciplinary Excellence Centre, Faculty of Medicine, University of Szeged, Semmelweis u. 6, H-6725 Szeged, Hungary

**Keywords:** oxidative stress, redox, antioxidant, multiple sclerosis, biomarker, neurodegenerative disease, personalized medicine

## Abstract

Worldwide, over 2.2 million people suffer from multiple sclerosis (MS), a multifactorial demyelinating disease of the central nervous system. MS is characterized by a wide range of motor, autonomic, and psychobehavioral symptoms, including depression, anxiety, and dementia. The blood, cerebrospinal fluid, and postmortem brain samples of MS patients provide evidence on the disturbance of reduction-oxidation (redox) homeostasis, such as the alterations of oxidative and antioxidative enzyme activities and the presence of degradation products. This review article discusses the components of redox homeostasis, including reactive chemical species, oxidative enzymes, antioxidative enzymes, and degradation products. The reactive chemical species cover frequently discussed reactive oxygen/nitrogen species, infrequently featured reactive chemicals such as sulfur, carbonyl, halogen, selenium, and nucleophilic species that potentially act as reductive, as well as pro-oxidative stressors. The antioxidative enzyme systems cover the nuclear factor erythroid-2-related factor 2 (NRF2)-Kelch-like ECH-associated protein 1 (KEAP1) signaling pathway. The NRF2 and other transcriptional factors potentially become a biomarker sensitive to the initial phase of oxidative stress. Altered components of the redox homeostasis in MS were discussed in search of a diagnostic, prognostic, predictive, and/or therapeutic biomarker. Finally, monitoring the battery of reactive chemical species, oxidative enzymes, antioxidative enzymes, and degradation products helps to evaluate the redox status of MS patients to expedite the building of personalized treatment plans for the sake of a better quality of life.

## 1. Introduction

Multiple sclerosis (MS) is an immune-mediated demyelinating disease of the brain and spinal cord, which over 2.2 million people worldwide suffer from; it affects primarily young adults from 20 to 40 years of age. After one to two decades, many MS patients enter a progressive phase of the disease. As survival rates have improved, MS patients suffer throughout the adult life. Years lived with disability begin to increase steeply early in the second decade of life and disability-adjusted life years peak in the sixth decade of life [1]. MS encompasses a wide range of symptoms from motor and autonomic dysfunctions to psychobehavioral disturbances, including gait difficulties, paresthesia, spasticity, vision problems, dizziness and vertigo, incontinence, constipation, sexual disturbances, pain, cognitive and emotional changes, anxiety, and depression [2,3,4,5]. Increased risk of depression and painful conditions in chronic illnesses are likely to be mediated by the kynurenine pathway of tryptophan metabolism [6,7].

Several genetic susceptibilities, environmental factors, and the aging process have been proposed to alter the risk of developing MS, but the underlying cause of the disease remains unknown [8,9]. The most typical pathomechanisms involved in MS are simultaneous inflammatory and neurodegenerative processes [10,11]. Main pathological findings of MS include the blood-brain barrier disruption, multifocal inflammation, demyelination, oligodendrocyte loss, reactive gliosis, and axonal degeneration [12]. Multifocal immune-mediated destruction of myelin and oligodendrocytes leading to progressive axonal loss is a main cause of neurological deficits in MS [13,14,15].

The diagnosis of MS is confirmed by the presence of two or more multifocal inflammatory or demyelinating attacks in the central nervous system (CNS) with objective clinical evidence, a single attack with magnetic resonance imaging (MRI)-detected lesions, and positive cerebrospinal fluid (CSF) analysis, or insidious neurological progression with positive brain MRI or CSF analysis [16]. The symptomatic course classifies MS into four subtypes. Approximately 85% of MS patients have alternating episodes of neurological disability and recovery that last for many years, termed relapsing-remitting MS (RRMS). Almost 90% of RRMS patients progress to steady neurological decline within 25 years, termed secondary progressive MS (SPMS). Nearly 10% of MS patients suffer from steady deterioration of neurological functions without recovery, termed primary progressive MS (PPMS). As few as 5% of MS patients present progressive neurological deficits with acute attacks with or without recovery, termed progressive-relapsing MS (PRMS) [17]. However, PRMS is no longer considered a subtype of MS and is now grouped into PPMS with active disease of new symptoms or changes in the MRI scan [18]. In addition, clinically isolated syndrome (CIS) is a single episode of monofocal or multifocal neurological deficits that lasts at least 24 h. CIS is one of the courses of MS disease [12].

As in other neurodegenerative diseases, MS is a clinically classified disease of CNS in which multifactorial factors, including genetic, environmental, socioeconomic, cultural, personal lifestyle, and aging play an initial role to form a causative complex, eventually converging into similar pathognomonic clinical pictures [4]. Inflammatory and demyelinating attacks are unique manifestations in MS, but different pathomechanisms govern the distinguished clinical courses in each subtype of MS.

Currently there is no cure for MS. Disease-modifying therapy is the mainstay of MS treatment. Immunomodulators, immunosuppressors, and cytotoxic agents are main groups of medicine. Immunomodulators, such as interferon (IFN)-beta (β), glatiramer acetate (GA), and siponimod are used for CIS. Short courses of high-dose corticosteroid methylprednisolone alleviate acute flare-ups of RRMS, indicating that an inflammatory process predominates in RRMS and relapse prevention of RRMS [19,20]. Siponimod is indicated for RRMS [20]. For active RRMS monoclonal antibodies alemtuzumab and ocrelizumab, and immunomodulator dimethyl fumarate, fingolimod, and teriflunomide are prescribed besides IFN-β and GA. For highly active RRMS, cytotoxic agents cladribine and mitoxantrone are indicated, besides immunomodulatory fingolimod, and monoclonal antibodies, natalizumab and ocrelizumab. PPMS and SPMS are characterized by a neurodegenerative process leading to neural death [18]. An immunosuppressive monoclonal antibody ocrelizumab is indicated for treatment of PPMS [21] and diroximel fumarate, siponimod, and ofatumumab were recently licensed for treatment of SPMS [20,22,23] (Table 1).

Either causative, accompanying, or resultant events of inflammation and neurodegeneration, disturbance of reduction-oxidation (redox) metabolism have been observed and play a crucial role in pathogenesis of MS [27,28,29]. The serum proteomics revealed that ceruloplasmin, clusterin, apolipoprotein E, and complement C3 were up-regulated in RRMS patients, compared to healthy controls. Vitamin D-binding protein showed a progressive trend of oxidation and the increased oxidation of apolipoprotein A-IV in progression from remission to relapse of MS [30]. CSF samples of patients in the remission stage of RRMS showed higher purine oxidation product uric acid, reduced antioxidant, and increased intrathecal synthesis of IgG [31]. The observations suggest the presence of redox metabolism disturbance and involvement of inflammatory process in RRMS. Furthermore, higher serum alpha (α)-tocopherol levels were associated with reduced T1 gadolinium (Gd^+^)-enhancing lesions and subsequent T2 lesions in MRI of RRMS patients on IFN-β. Antioxidant glutathione (GSH) mapping showed lower GSH concentrations in the frontoparietal region of patients suffering from PPMS and SPMS than RRMS and no significant difference between those of RRMS and controls. Thus, the oxidative stress in CNS was linked to neurodegeneration in progressive types of MS [32].

This review article reviewed participants of redox homeostasis, including reactive chemical species, oxidative and antioxidative enzymes, and resulting degradation products, all of which serve as evidence of biochemical assaults and tissue injuries under redox disequilibrium. Literature research was carried out using PubMed/MEDLINE and Google Scholar, using appropriate search terms, such as: “reactive oxygen species”, “oxidative stress”, “redox”, “biomarker”, “neurodegenerative disease”, “inflammation”, “neurodegeneration”, and/or “multiple sclerosis”. Meta-analyses, systematic reviews, expert reviews, case-control studies, and cohort studies were included in this review. Selection criteria and a risk of bias assessment are described in Table A1 and Table A2 of Appendix A. Alterations of redox components in MS in general, phase-, and treatment-specific components were reviewed in search of a potential diagnostic, prognostic, predictive, and/or therapeutic redox biomarker. Finally, monitoring different redox components, including oxidative enzymes, antioxidative enzymes, and degradation products during the disease progression helps to evaluate the redox status of MS patients, thus expediting the building of the most appropriate personalized treatment plans for MS patients [33]. 

## 2. Oxidative Stress

A redox reaction is a type of chemical reaction that involves the transfer of electrons between two molecules. A pair of electrons transfers from a nucleophile to an electrophile, forming a new covalent bond. The redox reaction is common and vital to the basic function of life such as cellular respiration, in which sugar is oxidized to release energy, which is stored in ATP. Redox metabolisms constitute multiple metabolic pathways involved in the series of redox chemical reactions indispensable for sustaining life and, at same time, engaged in removal of electrophilic oxidative species and other harmful nucleophiles. The dynamic activities range from a single electron transfer, enzyme reaction, chemical reaction cascade, to signaling in cells, tissues, organ systems, and whole organismal levels [34]. Oxidative stress is a state caused by an imbalance between the relative levels of production of reactive oxidizing metabolites and their elimination by the enzymatic or non-enzymatic antioxidant system. The oxidative state is induced by oxidative stressors either derived from xenobiotics or produced from the activities of oxidative enzymes and essential cellular constituents [35] (Figure 1a). Oxidative stressors are linked to the aging process, neurologic disease, and psychiatric disorders including Alzheimer’s disease (AD), Parkinson’s disease (PD), MS, amyotrophic lateral sclerosis (ALS), and depression [36,37,38,39,40].

### 2.1. Endogenous Oxidative Stressors: Oxidative Enzymes and Reactive Species 

Endogenous oxidative stressors are produced from cellular activities of cytosolic xanthine dehydrogenase (XDH), membrane-bound nicotinamide adenine dinucleotide phosphate (NADPH) oxidase, inflammatory lipoxygenase (LOX), and cyclooxygenase (COX), phagocytic myeloperoxidase (MPO) in respiratory burst, a second messenger nitrogen oxide (NO)-producing nitric oxide synthetase (NOS), mitochondrial and lysosomal electron transfer chain (ETC) enzymes, among others. Transition metals such as iron (Fe^2+^) and copper (Cu^+^) also play a crucial role in the formation of oxidative stressors via the Fenton reaction [41].

#### 2.1.1. Oxidative Enzymes Generating Reactive Oxygen Species

Reactive oxygen species (ROS) include several free radicals, such as superoxide (O_2_^−•^) and hydroxyl radical (OH^•^) and nonradical molecules such as hydrogen peroxide (H_2_O_2_) and organic hydroperoxide (ROOH) (Table 2). ROS are produced in endogenously in the cytosol, the plasma membrane, the membranes of mitochondria and endoplasmic reticulum, peroxisomes, and phagocytic cells [42,43] (Figure 2). ROS is difficult for direct measurement in biological tissues due to its highly reactivity and short life.

In the cytosol, ROS can be generated by soluble intercellular components, such as catecholamines, hydroquinones, flavins, and thiols (RSHs) that undergo reduction reactions [44]. The cytosolic enzyme XDH normally catalyzes xanthine, NAD^+^, and water (H_2_O) to urate, reduced form of NAD^+^, NADH, and hydrogen ion (H^+^). Reversible oxidation of cysteine residues or irreversible Ca^2+^-stimulated proteolysis converts XDH to xanthine oxidase (XO) that transfers electrons to molecular oxygen (O_2_), producing superoxide (O_2_^−•^) during xanthine or hypoxanthine oxidation [45]. The serum levels of uric acid, a major endogenous antioxidant was measured in patients with PPMM, RRMM, and SPMM. The uric acid levels were significantly lower in active MS than inactive MS, and the uric acid levels were independently correlated with gender, disease activity and duration of the disease [46] (Table 3).

The plasma membrane is a network of phospholipid bilayer and integral proteins which protects the cellular organelles from the outer environment and is responsible for several cellular functions, such as cell adhesion, ion transport, cell signaling, and phagocytosis. The main ROS of the plasma membrane is a superoxide (O_2_^−•^) produced by the membrane-bound enzyme nicotinamide adenine dinucleotide phosphate oxidase (NOX), which is composed of two membrane proteins, three cytosolic proteins, and a small GTP-binding protein [89,90]. The expression of NOX isoform NOX5 was significantly increased, but the expression NOX4 was significantly decreased in serum of RRMS patients, suggesting that differential NOX isoform expression contributes to OS-associated vascular changes in MS [48]. ROS is also produced by COX and LOX, which convert arachidonic acid to prostaglandins, thromboxanes, and leukotrienes. Phospholipase A_2_ generates ROS during arachidonic acid oxidation [91]. In the presence of transition metal ions, such as Fe^2+^ and Cu^+^, hydrogen peroxide (H_2_O_2_), organic hydroperoxide (ROOH), and organic peroxide (ROOR) produce hydroxyl (OH^•^), alkoxyl (RO^•^), and peroxyl radical (ROO^•^), respectively [92] (Table 3).

Superoxide dismutase (SOD) catalyzes the disproportionation of two superoxide (O_2_^−•^) into molecular oxygen (O_2_) and hydrogen peroxide (H_2_O_2_). These enzymes are present in almost all aerobic cells and in extracellular fluids. SODs contain metal ion cofactors that, depending on the isozyme, can be copper, zinc, manganese, or iron [93,94]. There are three isozymes in humans. Dimeric copper- and zinc-coordinated SOD1 is in the cytoplasm; tetrameric manganese-coordinated SOD2 is confined to the mitochondria; tetrameric copper- and zinc-coordinated SOD3 is extracellular [95]. The mitochondrial ROS production takes place at four protein complexes, ubiquinone, and cytochrome c of the ETC, embedded in the inner membrane of the mitochondria [96]. The Complexes I/III/IV utilize NADH as the substrate, while Complexes II/III/IV use succinic acid. The complex II is also glycerol 3-phosphate dependent. The primary mitochondrial ROS is superoxide (O_2_^−•^) that is converted by mitochondrial SOD into hydrogen peroxide (H_2_O_2_), which can be turned into hydroxyl (OH^•^) via the Fenton reaction [97]. The Complex I and III release superoxide (O_2_^−•^) into the mitochondrial matrix where it can damage the mitochondrial DNA, while the Complex III also releases superoxide (O_2_^−•^) into the intermembrane space where it is accessible to the cytosol [98]. Reduced levels of antioxidant α-tocopherol was observed in the blood of patients with Leber’s hereditary optic neuropathy, which is caused by mitochondrial mutation of the Complex I, suggesting that oxidative load was elevated and antioxidant capacity was compromised [99]. Other mitochondrial enzymes that contribute to hydrogen peroxide (H_2_O_2_) production are monoamine oxidases, dihydroorotate dehydrogenase, *α*-glycerophosphate dehydrogenase, and *α*-ketoglutarate dehydrogenase (*α*-KGDH) complex. Succinate dehydrogenase also generates ROS [100].

Significantly higher mean activity of SOD in erythrocyte lysates was reported in RRMS than controls. Interestingly, the SOD activity of CIS was higher than that of RRMS [101]. The erythrocyte SOD activity can be a diagnostic biomarker of CIS and RRMS. The SOD activity was significantly lower in the erythrocyte lysates of RRSM patients upon relapse than controls but increased following the intravenous administration of corticosteroid methylprednisolone and remained higher during remission period than controls. The SOD is a potent predictive and therapeutic biomarker. The serum/plasma samples also showed significantly higher mean SOD activity in RRMS compared to control groups [82]. However, platelet SOD1 and SOD2 activity was unchanged in MS patients [49]. Intriguingly, SOD activity was significantly low in CSF of CIS and RRMS patients, despite the significantly high activity of plasma SOD. Furthermore, there were negative correlations between the erythrocyte SOD activity and disease duration and expanded disability status scale (EDSS) in CIS and RRMS, between the erythrocyte SOD activity and Gd^+^ enhancement lesion volume in CIS patients [50]. These findings suggest the SOD activity as a possible diagnostic and prognostic marker (Table 3 and Table 4).

The peripheral blood mononuclear cells (PBMCs) SOD1 proteins and mRNA expression were significantly lower in RRMS patients than controls and became significantly more elevated following IFN-β1b treatment than the baseline [51]. These studies suggest SOD as a potential therapeutic biomarker (Table 4). However, the erythrocyte SOD activity remained unchanged following the treatment of natalizumab, a humanized monoclonal antibody against the cell adhesion molecule α4-integrin. But levels of carbonylated protein and oxidized guanosine were reduced [52].

In the inner membrane of mitochondria and the endoplasmic reticulum, a heme-containing monooxygenase cytochrome P450 (CYP) enzymes are responsible for oxidizing steroids, cholesterols, and fatty acids. The CYPs forms ROS superoxide (O_2_^−•^) and hydrogen peroxide (ROOH) by substrate cycling [102]. Protonation of hydrogen peroxide (ROOH) forms hydrogen peroxide (H_2_O_2_) which, furthermore, cleaves into hydroxy radicals (OH^•^). The redox cycling produces free radical semiquinone from quinoid substrates [103]. In the mitochondrial transport chain, flavoprotein reductase forms ROS by direct reduction of O_2_ and via the mediation of quinones. [104]. Superoxide (O_2_^−•^) is produced by XO in the reperfusion phase of ischemia, LOX, COX, and NADPH-dependent oxidase [105]. In the endoplasmic reticulum NADH cytochrome b5 reductase can leak electrons to molecular oxygen (O_2_) to generate superoxide (O_2_^−•^) during the NADPH-dependent oxidation of xenobiotics [106].

Most enzymes in the peroxisomes produce ROS during the catalysis of fatty acid *α*- and *β*-oxidation, amino acid and glyoxylate metabolism, and synthesis of lipidic compounds. A large fraction of hydrogen peroxide (H_2_O_2_) generated inside peroxisomes was observed to penetrate the peroxisomal membrane and diffuse to the surrounding media [107]. The peroxide can diffuse through the channel formed by the peroxisomal membrane protein Pxmp2 and hydrogen peroxide (H_2_O_2_) generated by the peroxisomal urate oxidase can release through crystalloid core tubules into the cytosol [108]. Meanwhile, peroxisomes also possess protective mechanisms to counteract oxidative stress and maintain redox balance. Reduction in peroxisomal gene and protein expression was observed in MS gray matter [109].

The lysosomal ETC plays a central role to support the positive proton gradient to maintain an optimal pH of the acid hydrolases [110]. The ETC is made up of a flavin-adenine dinucleotide, a b-type cytochrome and ubiquinone with the donor NADH and ending to acceptor molecular oxygen (O_2_), transferring three electrons. Superoxide (O_2_^−•^) is possibly produced in the acidic environment which favors dismutation of hydrogen peroxide (H_2_O_2_) into hydroxy radical (OH^•^) by ferrous iron [111]. Furthermore, ozone (O_3_) and ozone-like oxidants are generated from singlet oxygen (O_2_^1^Δ_g_) catalyzed by antibody or amino acid. Ozone (O_3_) reacts with superoxide anion (O_2_^−•^) to form hydrogen peroxide (H_2_O_2_) in the presence of Fe^+2^ [112].

#### 2.1.2. Oxidative Enzymes Generating Reactive Nitrogen Species

Reactive nitrogen species (RNS) are a group of nitrogen-congaing molecules including free radicals, nitric oxide (NO), and nitrogen dioxide (NO_2_). Free radicals are nitric oxide (NO^•^) and nitrogen dioxide (NO_2_^•^) radicals, while nonradicals are nitrite (NO_2_^−^) and nitrate (NO_3_^−^), among others (Table 2). RNS are derived from nitric oxide (NO) and superoxide anion (O_2_^−•^) produced by nitric oxide synthetase 2 (NOS2), NADPH oxidase, XO, LOX, and COX, among others [113,114]. At physiological concentrations, a gaseous molecule nitric oxide (NO) is a second messenger involved in blood pressure regulation, smooth muscle relaxation, defense mechanisms, immune regulation, and neurotransmission contributing the function of memory and learning [47].

Cross-sectional studies show that the levels of nitric oxide metabolites nitrite (NO_2_^−^) and nitrite (NO_3_^−^), measured as a total value (tNOx), are significantly higher in plasma or serum of patients with RRMS [83,115]. A longitudinal study revealed that higher serum tNOx is significantly correlated with relapsing rate, suggesting prognostic biomarker of NOS [116]. Many studies of CSF samples reported significantly higher levels of tNOx in RRMS and PPMS, compared to healthy controls [117,118]. A study observed significantly higher levels of CSF tNOx in RRMS than SPMS, suggesting an inflammatory role of RNS [119]. Significantly higher levels of CSF tNOx were reported in patients with acute relapsing phase of RRMS than those with stable remitting phase of RRMS [120,121] (Table 3 and Table 4).

In cGMP-dependent pathways, nitric oxide radical (NO^•^) generated by endothelial NOS in endothelium, brain, and heart relaxes blood vessels and maintains normal blood pressure, while nitric oxide radicals (NO^•^) produced by neuronal NOS serve as a neurotransmitter to regulate blood pressure in the brain. Inducible NOS (*i*NOS) in macrophages and smooth muscle cells gives rise to nitric oxide radicals (NO^•^) as in reaction to bacterial lipopolysaccharides and/or cytokines [122].

Nitric oxide radical (NO^•^) is produced from the metabolism of L-arginine by NOS that converts L-arginine into L-citrulline and nitric oxide radical (NO^•^) by a 5-electron oxidation of a guanidine nitrogen of L-arginine [123]. In mitochondria nitric oxide radicals (NO^•^) react with respiratory Complex III to inhibit electron transfer and facilitate superoxide anion (O_2_
^•−^) production. The nitric oxide radicals (NO^•^) also compete with molecular oxygen (O_2_) for the binding site at the binuclear center of cytochrome *c* oxidoreductase, inducing a reversible inhibition of cytochrome *c* oxidase. Nitric oxide (NO) neutralizes ROS [45]. However, RNS react with oxygen molecules (O_2_) and ROS, giving rise to a variety of nitrogen oxides (NOs), such as nitrogen dioxide radical (NO_2_^•^), nitrogen dioxide (NO_2_), dinitrogen trioxide (N_2_O_3_), peroxynitrite (ONOO^−^), nitrite (NO_2_^−^), and nitrate (NO_3_^−^). Higher concentrations of nitric oxide (NO) become toxic by forming nitrosothiols which oxidize tyrosine, cysteine, methionine, and GSH. In mitochondria, nitric oxide radicals (NO^•^) inhibit Complex I by *S*-nitrosation [124]. Together with other RNS, this contributes the damage of cell membranes, proteins, and lipid membrane leading to the degradation of mitochondria, lysosomes, and DNA. The chain of events culminates in the inhibition of immune response and production of carcinogenic nitrosamines [125]. Nitric oxide (NO) is also involved in metal homeostasis including Fe, Cu, and Zn [126]. 

Highly toxic peroxynitrite (ONOO^−^) is formed by the reaction of nitric oxide (NO) and superoxide anions (O_2_^−•^), leading to the production of more reactive compounds that oxidize methionine and tyrosine residues of proteins, lipids, and DNA. Reacting with superoxide anion (O_2_^−•^), nitric oxide radicals (NO^•^) form peroxynitrite which causes reversible inhibition of cellular respiration in the mitochondria [127]. In peroxisomes nitric oxide radicals (NO^•^) react with superoxide anions (O_2_^−•^) produced by XO to form peroxynitrite and hydrogen peroxides [111]. In addition, insulin resistance favors peroxintrite formation [128].

In response to bacterial lipopolysaccharides and inflammatory stimuli, *i*NOS generates nitric oxide (NO) that protects tissue hypoxia and serves as a neurotransmitter. However, overexpression *i*NOS, increase of nitric oxide (NO), and subsequent inflammation have been implicated in pathophysiology of neurodegenerative diseases including MS [47]. Calmodulin mediates the oxidative stress and inflammasome activation involving calmodulin-binding proteins, calcineurin, and calcium/calmodulin-dependent kinase II. Calmodulin inhibitors improved cognitive functions in an animal model of vascular dementia [129]. The *i*NOS activity is upregulated in acute MS plaques [19,53]. Increased activity and expression of *i*NOS in lymphocytes were found in active relapsing phase of RRMS [54]. CSF *i*NOS expression was shown in MS patients and mean CSF NOS activity was significantly higher, compared to controls [55] (Table 3).

#### 2.1.3. Reactive Sulfur Species

Reactive sulfur species (RSS) are sulfur-based redox-active compounds able to oxidize or reduce biomolecules under physiological conditions, often formed by thiols (RSHs) and disulfides (RSSHs). RSS include cysteine and methionine, GSH, trypanothione, and mycothiol [130] (Table 2).

Thiyl radicals (RS^•^), very reactive oxidants produced in the active site of enzymes such as the ribonucleotide reductase can react with nitric oxide radicals (NO^•^) [131]. Thiolate ions (RS^−^) are better nucleophiles than alkoxides because sulfur is more polarizable than oxygen [132]. Thiol (RSH) is a metal ligand [133]. Hydrogen sulfide (H_2_S) is produced from from L-cysteine by cistationine-γ-lyase. Hydrogen sulfide (H_2_S) increases the activity of *N*-methyl-D-aspartate receptor and β-adrenergic receptors through a cAMP-dependent protein kinase and activates NOS and the hemoxygenase favoring the formation of Nitric oxide (NO) and carbon monoxide (CO) from heme metabolism [134]. Hydrogen sulfide (H_2_S) is a metal ligand that reacts with other biological electrophilic sulfur species such as hydropersulfide (RSSR) and sulfenic acid (RSOH) [135]. Cysteine residues in GSH were found to be readily oxidized by superoxide anions (O_2_^−•^) to form singlet oxygen (O_2_^1^Δ_g_), glutathione disulfide (GSSG), and glutathione sulfonate (GSO_3_^−^) in a reaction involved with peroxysulphenyl radical (RSOO^•^). This mechanism may apply to cysteine residues in proteins [136]. Hydroxyl radicals (OH^•^) may also initiate the conversion of amino acids to peroxyl radicals [92]. Another reaction catalyzed by XO is the decomposition of S-nitrosothiols (RSNO), a RNS, to nitric oxide (NO), which reacts with a superoxide (O_2_^−•^) anion to form peroxynitrite (ONOO^−^) under aerobic conditions [137]. Hydropersulfide (RSSH) is a nucleophile, as well as electrophilic molecule that is readily reduced to extremely potent reductant thiol (RSH). Disulfide (RSSR) is electrophilic RSS that can be reduced to thiol (RSH). Hydropolysulfide (RSS_n_H) and dialkyl polysulfide (RSS_n_R) are like hydropersulfide (RSSH) [135]. RSS can interact with ROS, generating sulfur oxides such as peroxysulphenyl radical (RSOO^•^), sulfenate (RSO^−^), sulfinate (RSO_2_^−^), sulfonate (RSO_3_^−^), thiosulmonate (S_2_O_3_^2−^), and SO_4_^2 −^ [138]. Sulfenate (RSO^−^) reacts with other thiols to give disulfides, RSSR. RSS can also interact with RNS, leading to the formation of S-N hybrid molecules, such as thiazate (NSO^−^), thionitrite (SNO^−^) isomers, S-nitrosothiols (RSNOs), nitrosopersulfide (SSNO^−^), and the dinitrosylated sulfite adduct, SULFI/NO. S-nitrosothiols (RSNOs) can be reduced to thiol (RSH) and nitroxyl (HNO) [139]. XO catalyzes S-nitrosothiols (RSNOs) to nitric oxide (NO), which reacts with a superoxide (O_2_^−•^) anion to form peroxynitrite (ONOO^−^) under aerobic conditions [123]. The properties of thiazate (NSO^−^), thionitrite (SNO^−^) isomers, nitrosopersulfide (SSNO^−^) and polysulfides dinitrososulfites (Sulfi/NO), are to be determined and they appear to be a source of nitric oxide (NO) and nitroxyl (HNO) [140,141,142] (Table 2).

An antioxidant *N*-acetyl cysteine administration was reported to improve cognitive functions in patients in MS and *N*-acetyl cysteine supplement is under clinical trial for the treatment of fatigue in MS patients [143,144]. The levels of methionine reported mixed results. The plasma methionine levels were significantly reduced in RRMS and dietary methionine supplement was proposed for the treatment [145]. The level of methionine sulfoxide was elevated more than two-fold in CSF of MM patients and the reduction of dietary methionine was reported to slow the onset and progression of MM [146,147]. The levels of GSH in MS have not reached a consensus [147,148,149]. Trypanothione and mycothiol have not been investigated in MS. The serum S-nitrosothiol levels were increased in RRMS and SPMS, and selectively correlated with spinal cord injury and, thus, a high level of S-nitrosothiol is proposed to be a potential prognostic biomarker for spinal cord injury in MS [150] (Table 3 and Table 4).

#### 2.1.4. Reactive Carbonyl/Halogen/Selenium Species

Reactive carbonyl species (RCS) are metabolically generated highly reactive molecules with aldehydes and electronically excited (^3^L=O*) triplet carbonyls, known for their harmful reactions to nucleic acids, proteins, and lipids [92]. In addition, RCS are considered to participate in electrophilic signaling of adaptive cell response and post-transcriptional protein modification [151]. RCS are classified into α,β-unsaturated aldehydes; keto-aldehyde and di-aldehydes. In the presence of catalase and bicarbonate, XO was found to produce the strong one-electron oxidant carbonate radical anion from oxidation with acetaldehyde. The carbonate radical was likely produced in one of the enzyme’s redox centers with a peroxymonocarbonate intermediate [45]. Self-reaction of lipid peroxyl radical (LOO^•^) produced by the oxidation of polyunsaturated fatty acids (PUFAs) by hydroxyl radical (HO^•^) generates electronically excited triplet carbonyls (^3^L=O*) yielding to singlet oxygen (O_2_^1^Δ_g_) [89] (Table 2). RCS react with amines and thiols leading to advanced glycation end products (AGEs), biomarkers of ageing and degenerative diseases [152].

MPO, a lysosomal heme-containing enzyme present in granulocytes and monocytes catalyzes the conversion of hydrogen peroxide (H_2_O_2_) and chloride anion (Cl^−^) to hypochlorous acid (HClO) during the respiratory burst. An adipocyte producing hormone leptin stimulates the oxidative burst [153]. MPO also oxidizes tyrosine to tyrosyl radical [154]. MPO mediates protein nitrosylation, forming 3-chlorotyrosine (3-Cl-Tyr) and dityrosine crosslinks [155,156] (Table 2).

Studies on MPO activity reported mixed results. The mean MPO activity of peripheral leukocyte was observed reduced in a mixed population of MS patients, compared to controls [157]. Significantly higher serum MPO activity was measured in opticospinal phenotype (OSMS) of RRMS at relapse and remission and in conventional phenotype of RRMS at remission, compared to controls. A positive correlation was associated between Kurtzke’s EDSS and MPO activity at remission of OSMS [158] (Table 4). The mean MPO activity of peripheral leukocytes was found higher, but statistically not significant in RRMS, compared to controls [56] (Table 3). No study regarding CSF MPO activity in MS was found. Selenium is an essential micronutrient with similar chemical and physical properties to sulfur, but more easily oxidized and kinetically more labile than sulfur. Selenium is a component of proteinogenic selenocysteine, naturally occurring selenomethionine and selenoproteins, such as glutathione peroxidase (GPx), thioredoxin reductase (TRXR), and selenoprotein P [57]. Reactive selenium species (RSeS) are selenium-containing inorganic and organic compounds including selenite (O_3_Se^−2^), selenocysteine (C_3_H_7_NO_2_Se), and selenomethionine (C_5_H_11_NO_2_Se) [58] (Table 2). Selenium have both beneficial and harmful actions. At low concentration it works as an antioxidant, inhibiting lipid peroxidation and detoxifying ROS as a component of GPx and TRXR, while at high concentration it becomes a toxic pro-oxidant, generating ROS, inducing lipid oxidation and forming cross-linking in thioproteins [159] (Table 2). The serum selenium levels were measured significantly lower in MS patients, compared to controls, suggesting antioxidant capacity is impaired in MS [160] (Table 3).

#### 2.1.5. Exogenous Oxidative Factors

Oxidative stressors are generated in reaction to exogenous stimuli such as pollutants, food and alcohol, cigarette smoke, heavy metals, chemotherapy, drug and xenobiotics, or radiation. Aging becomes more susceptible to their insults. Organic solvents, organic compounds such as quinone, pesticides and heavy metals including lead, arsenic, mercury, chromium, and cadmium are common sources of oxidative stressors [41]. Ultraviolet and infrared-B radiations generate oxygen radicals endogenously [161] (Figure 1a). The levels of serum arsenic, malondialdehyde (MDA), and lactate were elevated and ferric-reducing activity of plasma was reduced RRMS patients and the levels of serum lithium were significantly lower and the levels of nitric oxide (NO) were higher in RRMS patients, compared to healthy controls, suggesting environmental factors seem to play a role in pathogenesis of MS [162,163].

## 3. Reductive Stress

### 3.1. Reactive Nucleophilic Species

Endogenous reductive stressors include nucleophilic free radical, inorganic, and organic molecules and antioxidative enzyme. (Figure 1b). Superoxide (O_2_^−•^) anion is one of the reactive nucleophilic species and a powerful reducing agent under physiological conditions, which initiates reaction cascades generating another ROS, such as hydrogen peroxide (H_2_O_2_) and sulfur dioxide (SO_2_) derivatives. Hydrogen sulfide (H_2_S), thiolate (RS^−^), hydropersulfide (RSS^−^), and disulfide (RSSR) are reactive nucleophilic species that can participate in nucleophilic substitution in vivo [133]. Selenium is more nucleophilic than sulfur due to its greater electron density. The selenol (RSeH) portion of selenocysteine (C_3_H_7_NO_2_Se) is ionized at physiological pH, making it more nucleophilic against oxidative species [164,165] (Table 2). 

### 3.2. Antioxidative Enzymes

Reductive stress is induced by excessive levels of reductive stressors that results from an elevation in GSH/GSSG ratio, NAD^+^/NADH, NADP^+^/NADPH and/or or overexpression of antioxidative enzymatic systems such as the GSH system, catalase, thioredoxin-peroxiredoxin (TRX-PRDX) system, α-ketoglutarate dehydrogenase (GPDH), and glycerol phosphate dehydrogenase [166,167]. The reductive stressors deplete reactive oxidative species and are harmful as oxidative stressors and implicated in pathological processes in AD, PD, and sporadic motor neuron disease, among others [168]. 

The GSH system consists of GSH, the enzymes for synthesis and recycling including gamma-glutamate cysteine ligase, glutathione synthetase, glutathione reductase (GSR), and gamma glutamyl transpeptidase, and the enzymes for metabolism and antioxidation including glutathione S-transferase and GPx [169]. The GPx is an enzyme containing four selenium-cofactors that catalyzes the reduction of hydrogen peroxide (H_2_O_2_) to water molecule (H_2_O) and organic hydroperoxide (ROOH) to alcohol (ROH) by converting reduced monomeric GSH to GSSG. Glutathione s-transferases show high activity with lipid peroxides [170]. Eight isozymes are in the cytosol, membrane, and plasma, protecting the organisms from oxidative stress [171].

Most studies on peripheral blood GPx activity reported nonsignificant results in a mixed population of MS [56,59,172,173,174]. However, lower mean GPx activity of erythrocyte lysates in remission and higher mean GPx were reported in acute relapse of RRMS [60]. GPx activity in CSF was found lower in MS patients [61]. The GSR activity of lymphocyte and granulocyte lysates were not significantly different in MS, compared to controls. However, a significant correlation of GPx and GRx was observed in controls, but not in MS [62]. Mean GRx activity of CSF was found significantly higher in MS patients [61] (Table 3 and Table 4).

Catalases a tetrameric heme- or manganese-containing dismutase that catalyzes the conversion of two hydrogen peroxide (H_2_O_2_) molecules to water (H_2_O) in the presence of small amount of hydrogen peroxide. The cofactor is oxidized by one molecule of hydrogen peroxide and then regenerated by transferring the bound oxygen to a second molecule of substrate. The enzyme is located in the peroxisomes, the cytosol of erythrocytes, and the mitochondria, removing harmful hydrogen peroxides to prevent cellular and tissue damage [175].

Studies on the catalase activity of peripheral blood samples reported equivocal results in MS. The catalase activity of granulocyte lysates was found lower in MS patients, compared to controls [176]. The activities of CSF and plasma catalase were found increased in CIS and RRMS patients, compared to healthy controls, and MS patients with lower EDSS had higher plasma and CSF catalase activities [84] (Table 3 and Table 4).

In TRX-PRDX system PRDXs catalyze the reduction of H_2_O_2_ to H_2_O. H_2_O_2_ oxidizes the peroxidatic cysteine of PRDXs to protein sulfenic acid (PSOH), which can react with the thiol (SH) group of the resolving cysteine to yield the formation of an inter-(typical) or intramolecular (atypical) disulfide bond. TRX/TRXR system mediates the reduction of the PRDX disulfide bond. TRX reduced state is maintained by the flavoenzyme TRXR in the presence of NADPH. When H_2_O_2_ exceeds the normal levels, PRDXs are overoxidized from PSOH to protein sulfinic acids (PSO_2_H). The latter can be reduced back to the native form of the enzyme by sulfiredoxin (SRX) in the presence of ATP. However, further oxidation of PRDXs to PSO_3_H is irreversible [177].

Serum Trx1 was significantly increased in the newly diagnosed MS patients, compared to controls. *TRX1* and *APEX1* mRNA expressions were significantly higher in the newly diagnosed MS patients, patients under INF-β treatment, and patients who received immunosuppressant azathioprine or betamethasone, compared to healthy controls [178]. *PRDX2* mRNA is upregulated and PRDX2 expression is higher in MS lesions white matter of autopsy tissue of patients its expression level is positively correlated with the degree of inflammation and oxidative stress [179] (Table 3 and Table 4).

*α*-KGDH is a mitochondrial enzyme in Krebs cycle, which catalyzes α-ketoglutarate, coenzyme A and NAD^+^ to succinyl-CoA, NADH and CO_2,_ transferring an electron to the respiratory chain [180]. KGNH activity is sensitive to redox status. H_2_O_2_ reversibly inhibits KGNH by glutathionylation of lipoic acid cofactor, resulting reducing electron supply to the respiratory chain. A lipid peroxidation product 4-hydroxy-2-nonenal (4-HNE) reacts with lipoic acid cofactor, inhibiting α-KGDH activity [181]. The pyruvate tolerance test showed higher activity of α-KGDH in serum of MS patients [182]. However, reduced expression and activity of mitochondrial *α*-KGDH was observed in demyelinated axons that correlated with signs of axonal dysfunction (Table 3) [183].

α-GPDH catalyzes the reversible redox conversion of dihydroxyacetone phosphate to sn-glycerol 3-phosphate, linking carbohydrate and lipid metabolism. A loss of α-GPDH in oligodendrocytes were observed in chronic plaques of MS patients, suggesting the presence of antioxidant capacity impairment [63] (Table 3).

Nrf2 is a transcriptional factor of the antioxidative enzyme genes including catalase, GPx, GRx, glutathione S-transferase, and SOD. In response to oxidative stress, the Kelch-like ECH-associated protein 1 (KEAP1) inhibits the ubiquitin-proteasome system in the cytosol and facilitates the translocation of Nrf2 into the nucleus to bind to the *cis*-acting enhancer sequence of the promotor region, the antioxidant response elements [184,185]. Activation of the Nrf2-Keap1 pathway has been observed in various types of cancers, accompanied with reduced antioxidant capacity and elevated oxidative stress and inflammation [186]. The cytoplasmic and nucleic Nrf2 protein expression of PBMC was increased and correlated with clinical improvement in MS patients on 14-month course of natalizumab, an α4 integrin receptor blocker [110] (Table 3 and Table 4).

Other transcriptional factors involved in energy metabolism have been investigated. Peroxisome proliferator-activated receptors (PPARs) are a transcriptional factor of the gene regulating energy metabolism including glucose metabolism, fatty acid oxidation, thermogenesis, lipid metabolism, and anti-inflammatory response [187]. PPARs have attracted growing attention as promising targets of many diseases such as diabetes and hyperlipidemia [188]. An isoform PPAR-gamma (PPAR-γ) was elevated in CSF samples of MS, compared to controls [64]. Peroxisome proliferator-activated receptor gamma coactivator 1-α (PGC-1α) 4 integrin receptor blocker is a transcriptional coactivator that regulates the genes involved in energy metabolism. Reduced PGC-1α expression was associated with mitochondria changes and correlated with neural loss in MS [189] (Table 3 and Table 4). 

### 3.3. Exogenous and Endogenous Antioxidants

A daily diet rich in naturally occurring polyphenolic antioxidants, such as flavonoids and phenolic acids are regularly recommended for disease prevention and antioxidant supplements, such as vitamin C, vitamin E, *N*-acetyl cysteine, L-carnitine and folic acid are frequently employed as a complementary therapy for various diseases [190,191]. Those preventive and therapeutic measures are based on the pathogenesis of diseases which are induced and developed under oxidative cellular environment. Furthermore, plant polyphenols were reported to alleviate oxidative stress and enhance neuroprotection [192]. Olive leaf-derived polyphenols are strong antioxidants and their therapeutic use are of particular interest against oxidant-induced diseases including cancer and neurodegenerative diseases [193,194,195]. The Mediterranean diet can present a biphasic dose-response curve toward hormetic stimuli, preventing low-grade inflammation and inflammageing [196,197]. There is a growing interest in supplementation of nonessential compounds that trigger the redox feedback loop activating the nucleophilic response, parahormesis (Figure 1c) [198].

N-Methyl-D-aspartic acid receptor antagonist memantine and memantine-ferulic acid conjugate improved oxidative stress in patients with AD [199]. A meta-analysis of randomized controlled trials showed that an exogenous antioxidant *N*-acetylcysteine supplement improved cognitive function in patients with schizophrenia [200]. Traditional Chinese medicine curcumin is an antioxidant and anti-inflammatory molecule that relieves pain and stress at least partly through the kynurenine metabolic pathway of tryptophan metabolism [201]. The kynurenine pathway produces endogenous oxidative and antioxidative metabolites. Single nucleotide polymorphism of the first rate-limiting enzyme indoleamine 2,3-dioxygenase 1 was associated with inflammation and depressive symptoms and influenced the age onset of neurodegenerative diseases [202]. The elevated levels of an oxidative metabolite 3-hydroxykynurenine and an antioxidative kynurenic acid were linked to depressive symptom of patients with stroke [203]. Meanwhile, administration of kynurenic acid induced antidepressant-like effects in animal model of depression [204]. 

However, unmonitored chronic antioxidant supplementation imposes reductive stress, the counterpart of oxidative stress. The reductive stress-induced inflammation is observed in hypertrophic cardiomyopathy, muscular dystrophy, pulmonary hypertension, rheumatoid arthritis, AD, and metabolic syndrome [168]. In adipose tissue a long-term antioxidant supplementation caused a paradoxical increase in oxidative stress which was associated with mitochondrial dysfunction [205]. Leptin secreted from adipose tissue serves as an inflammatory mediator and subsequent development of leptin resistance make obese individuals more susceptible to autoimmune disease including MS [206] (Figure 1c). 

Vitamin supplements are recommended for the treatment of MS, as nutritional deficits are frequently observed in patients with MS [207]. MS induced by reductive stress has not been reported, but it deserves to monitor redox status in MS patients.

## 4. Degradation Products under Oxidative Stress 

### 4.1. Proteins

Protein carbonyls are degradation products of reactions between reactive species and proteins, resulting in a loss of function or aggregation. Quantification with 2,4-Dinitrophenylhydrazine products showed increased carbonylation in plasma and serum of RRMS patients [65,66,67]. The plasma carbonyl levels were elevated in SPMS and correlated with the EDSS and the Beck Depression Inventory [208]. The levels of carbonyl groups were elevated in serum of patients with RRMS and lowered in the group of RRMS patients treated with INF-β [74]. The levels of CSF carbonyl proteins measured were elevated in RRMS and progressive MS [209,210] (Table 3 and Table 4).

A highly active RNS reacts with tyrosine residues of proteins to form nitrotyrosines, leading to the alternation of protein conformation function. The main product of tyrosine oxidation is 3-nitrotyrosine (3-NO-Tyr), formed by the substitution of a hydrogen by a nitro group in the phenolic ring of the tyrosine residues. The content of 3-NO-Tyr is assessed by western blotting, high-performance liquid chromatography (HPLC), gas chromatography-mass spectrometry (GC/MS), and enzyme-linked immunosorbent assay (ELISA) [72]. Mean 3-NO-Tyr was observed significantly higher in plasma and serum of RRMS and SPMS patients and significantly higher 3-NO-Tyr was found in SPMS than RRMS [66,68,69]. Decreased mean 3-NO-Tyr was reported following a relapse and corticosteroid treatment [70]. In the serum of MS patients, 3-NO-Tyr was found significantly lower following INF-β1b treatment [71]. Following GA treatment, 3-NO-Tyr was found significantly reduced in peripheral leukocytes [69] (Table 3 and Table 4).

Protein glutathionylation is a redox-dependent posttranslational modification that results in the formation of a mixed disulfide between GSH and the thiol group of a protein cysteine residue [211]. Protein glutathionylation is observed in response to oxidative or nitrosative stress and is redox-dependent, being readily reversible under reducing conditions. Extracellular SOD, α1-antitrypsin and phospholipid transfer protein were found glutathionylated at cysteine residues in CSF of MS Patients, witnessing the footprints of oxidative assault of MS [210].

Oxidative environments generate oxidized tyrosine orthologues such as o-tyrosine, m-tyrosine, nitrotyrosine, and dityrosine. Dityrosine was elevated in serum of RRMS patients [73,74]. Advanced oxidation protein products (AOPPs) are uremic toxins produced in reaction of plasma proteins with chlorinated oxidants such as chloramines and hypochlorous acid (HClO) [212]. The levels of AOPPs were significantly higher in plasma of MM patients [213]. The levels of AOPPs were significantly higher in plasma and CSF of CIS and RRMS patients than healthy controls, and the AOPPs levels were significantly higher CIS than RRMS. Furthermore, the levels of AOPPs were significantly higher in patients with higher EDSS scores than lower ones [50]. The AOPPs levels were decreased in serum of RRMS patients treated with IFN-β [74] (Table 3 and Table 4).

AGEs are a group of glycotoxins produced in reaction of free amino groups of proteins, lipids, or nucleic acids and carbonyl groups of reducing sugars. The AGEs can accumulate in tissues and body fluids, resulting in protein malfunctions, reactive chemical production, and inflammation [214]. The levels of AGEs were significantly elevated in serum of RRMS patients, but no significant change was observed after IFN-β treatment [74]. The concentrations of AGEs were significantly higher in brain samples of MS patients, compared to nondemented counterparts. The levels of free AGEs were correlated in CSF and plasma samples of MS patients, but not protein-bound AGEs [75] (Table 3 and Table 4).

### 4.2. Amino Acids

Asymmetric dimethylarginine (ADMA) is a L-arginine analogue produced in the cytoplasm in the process of protein modification. The formation of ADMA is dependent on oxidative stress status. ADMA is elevated by native or oxidized LDL and interferes with L-arginine in the production of nitric oxide (NO) [215]. Significantly higher ADMA concentrations were observed in serum and CSF of patients with RRMS and SPMS, while levels of arginine, L-homoarginine, nitrate, nitrite, ADMA did not differ between patients with MS and healthy controls [76] (Table 3 and Table 4).

### 4.3. Lipid Membrane and Lipoproteins 

Lipids in biological membrane are major target of OS. Peroxidation of lipid membrane is initiated by ROS, including superoxide anion (O_2_^−^^•^), hydroxyl radical (OH^•^), hydrogen peroxide (H_2_O_2_), and singlet oxygen (O_2_^1^Δ_g_); and by RNS, including nitric oxide radical (NO^•^), peroxynitrite (ONOO^−^), and nitrite (NO_2_^−^) stealing electron from PUFAs, such as arachidonic (20:4) and docosahexaenoic acid (22:6) [216]. The abstraction of *bis*-allylic hydrogen of PUFA leads to the formation of arachidonic acid hydroperoxyl radical (ROO^•^) and hydroperoxide (ROOH) in a chain reaction manner [45]. A portion of arachidonic acid peroxides and peroxy radicals generate endoperoxides rather than hydroperoxide (ROOH). The endoperoxides undergoes subsequent formation of a range of bioactive intermediates, such as F2-isoprostanes (F2-isoPs), MDA and 4-HNE. Hexanoyl-lysine (HEL) adduct is a lipid peroxidation by-product which is formed by the oxidation of omega-6 unsaturated fatty acid, such as linoleic acid [217]. Hydroxyoctadecadienoic acid (HODE) is derived from the oxidation of linoleates, the most abundant PUFAs in vivo [73,211]. Meanwhile, a cyclic sugar compound inositol is a major antioxidant component of the lipid membrane, which scavenges reactive species [54]. 

Studies on blood F2-isoPs levels reported increases in MS, especially in RRMS and SPMS subtypes compared to controls [77,78]. A study on CSF F2-isoPs levels presented three times higher in patients with MS than ones with other neurologic diseases [79]. The levels of F2-isoPs were moderately correlated with the degree of disability, suggesting a role as a prognostic marker [80]. MDA is highly reactive aldehyde generated by the reaction between reactive species and polyunsaturated lipids to form adducts with protein or DNA [81]. Studies on blood or serum MDA reported higher levels in MS patients [83,166]. The blood MDA levels were significantly higher in RRMS than controls or CIS, higher in RRMS than in remission, and higher in remission than controls. MDA levels were elevated at relapse, while lowered at day 5 of corticosteroid treatment [82,149]. Studies quantifying CSF MDA consistently reported higher levels in CIS and RRMS than controls [61,82,85,177]. There are positive correlations between MDA levels of plasma and CSF, and MDA levels in plasma/CSF and EDSS [84]. The levels of 4-HNE were elevated in the CSF of PPMS, RRMS, and SPMS patients, particularly in PPMS [86]. No study regarding HEL in MS was found in literature search. The serum 13-HODE was identified as a part of metabolomic signatures associated with more severe disease such as non-relapse-free MS or MS with higher EDSS [87]. The levels of 9-HODE and 13-HODE were significantly increased in CSF of CIS and RRMS patients, compared to healthy controls, but baseline levels of HODE did not differ between patients with signs of disease activity during up to four years of follow-up and patients without MS [218] (Table 3 and Table 4).

Cholesterol oxidization products oxysterols were studied. Levels of plasma oxysterols increased in progressive MS patients and oxysterol levels were positively correlated with apolipoprotein C-II and apolipoprotein E. Furthermore, oxysterol and apolipoprotein changes were associated with conversion to SPMS [88]. Increased levels of oxidized low-density lipoprotein (oxLDL) in the serum and higher serum levels of autoantibodies against oxLDL were reported in MS patients [219,220]. Although studies on HDL levels in MS patients reported mixed results, lowered HDL antioxidant function in MS patients was observed, suggesting the involvement of lipoprotein function MS pathogenesis [219,220,221,222]. In mixed population of MS, decreased serum 24S-hydroxycholesterol and 27-hydroxycholesterol and increased CSF lathosterol, compared to healthy controls [223] (Table 3 and Table 4).

### 4.4. Nucleic Acid

The biomarkers of oxidative damage of nucleic acids—8-Hydroxy-2′-deoxyguanosine (8-OH2dG) and 8-hydroxyguanosine (8-OHG)—can be assessed by ELISA, as well as by direct methods, such as HPLC and GC/MS [224,225]. Elevated levels of 8-OH2dG were reported in the blood of RRMS patients. DNA oxidation products were proposed as diagnostic biomarkers for MS [152] (Table 3 and Table 4).

## 5. Conclusion and Future Perspectives

Redox biomarkers are classified by original cellular components and enzyme mechanisms of action in redox homeostasis. Redox status can be assessed by the measurement of reactive chemical species, oxidative or antioxidative enzyme activity, and degradation products derived from proteins, amino acids, lipid membrane, and nucleic acids. Measurement of reactive chemical species and oxidative enzyme activities assesses intensity of oxidative stress, while measurement of antioxidative activity analyses compensatory capacity. Fine measurement of the various redox components may reveal diagnostic, prognostic, or predicative value to differentiate the disease status and progression (Figure 3).

The levels of nitric oxide (NO) metabolites, *S*-nitrosothiol, and the activities of oxidative enzymes including SOD, MPO, and *i*NOS have been found significantly different to patients with MS, compared to healthy controls. The levels of antioxidants including uric acid and selenium, activities of antioxidative enzymes including GSR, catalase, and TRX-PRDX, and concentrations of transcriptional factors Nrf2, PPARs, and PGC-1α have been found significantly changed in MS patients. The protein degradation products including protein carbonyls, 3-NO-Tyr, glutathionylation, AOPPs, and AGEs, an amino acid by-product ADMA, the lipid and cholesterol degradation products including F2-isoP, MDA, 4-HNE, HODE, oxocholesterols, and oxLDL, and the nucleic acid degradation product 8-OH2dG significantly increased in samples of MS patients. Thus, the oxidative enzymes, antioxidative enzymes, and redox degradation products have been identified as promising biomarkers for the diagnosis of MS. Furthermore, SOD, 3-NO-Tyr, and MDA are sensitive to subtypes of MS, CIS and RRMS, RRMS and SPMS, RRMS and remission, respectively. tNOx, *S*-nitrosothiol, SOD, MPO, GSR, catalase, protein carbonyls, AOPPs, F2-isoP, MDA, and oxycholesterols were correlated with EDSS and thus they are potential prognostic biomarkers for MS. SOD, Nrf2, protein carbonyls, 3-NO-Tyr, AOPPs, and MDA were observed sensitive to the treatment of MS, being possible predictive biomarkers. Finally, SOD is a possible drug target of MS as a therapeutic marker (Table 3).

In addition to biomarkers described above, the search for novel biomarkers has become a growing interest in neurodegenerative diseases. A micro RNA (miRNA) is a short non-coding RNA molecule consisting of approximately 22 nucleotides, which functions in posttranscriptional gene silencing. miRNAs have been linked to pathogenesis of various diseases including cancer, autoimmune diseases, and neurodegenerative disease such as PD [226]. Dysregulated interactions of miRNAs have been reported in mild cognitive impairment and AD. The associated genes were related to regulation of ageing and mitochondria [227]. Dysregulations of various miRNAs have been observed in AD, PD, Huntington’s disease, and ALS and thus miRNAs were proposed to be potential diagnostic and therapeutic biomarkers of neurodegenerative diseases [228]. The link between environmental factors and miRNA dysregulations in MS was discussed [229]. Furthermore, miRNAs in blood and CSF samples of patients with MS as diagnostic and prognostic biomarkers have been reviewed recently [230].

Long Interspersed Nuclear Element-1 (LINE-1) is an autonomous non-long terminal repeat retrotransposon that creates genomic insertions through an RNA intermediate. The increased number of germline and somatic LINE-1s have been linked to the risk and progression of cancer as well as neurodegenerative and psychiatric diseases. An increased burden of highly active retrotranposition competent LINE-1s have been associated with the risk and progression of PD and LINE-1s were proposed as possible therapeutic biomarkers that can be targeted by reverse transcriptase [231]. Furthermore, LINE-1s was considered involved in irregular immune response and participate in pathogenesis of MS [232].

A search for demographic correlation between single nucleotide polymorphisms and MS has been under extensive study. Inflammation-mediating chemokine receptor V Δ32 deletion was not found correlated with MS [233]. Large-scale genome projects such as genome-wide association (GWA) studies generated polygenic risk scores for prediction of risk and progression of multifactorial neurodegenerative diseases. Large-scale pathway specific-genetic risk profiling expedited redox-related biological pathways to identify causal genes and potential therapeutic targets [234]. One of future challenges is a search for correlations with uncatalogued structural variants in MS.

Considering the dynamics of redox homeostasis, the amounts of reactive species, activities of oxidative and antioxidative enzymes, and concentrations of degradation products presumably differ during the progression of MS. The early phase presents an elevation of oxidative enzyme activity and a subsequent elevation of the activity of counteracting antioxidative enzymes with unchanged levels of degradation products. As the activity of antioxidative enzymes becomes compromised due to increasing oxidative stress, the amount of degradation products gradually increases, while the antioxidative enzyme activities slowly wane and fatigue. Eventually, the antioxidative response exhausts with elevated activities of oxidative enzymes and elevated levels of degradation products (Figure 4). Further studies and exploration into novel biomarkers are expected in search of a robust battery of biomarkers indicative to the redox status, in order to realize a fine calibration of major redox components that helps identify the disturbance of redox homeostasis, restore the nucleophilic tone and the most importantly build the best personalized treatment of MS for the sake of better quality of life [235].


## Figures and Tables

**Figure 1 biomedicines-08-00406-f001:**
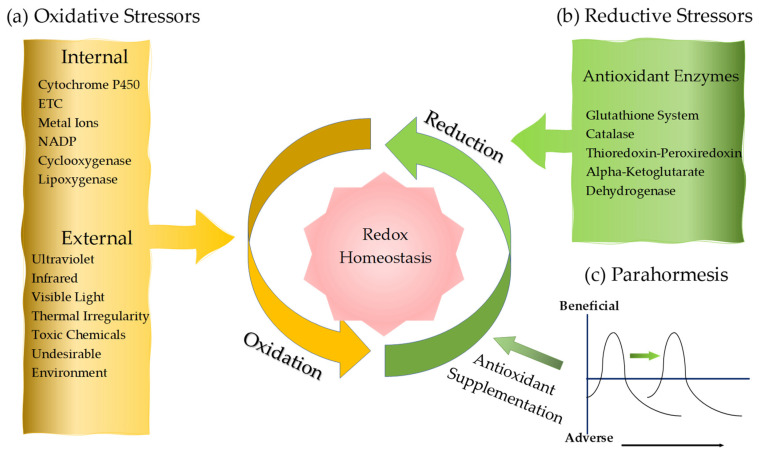
Redox homeostasis and antioxidant supplementation. (**a**) Oxidative stressors comprise of external and internal stressors, exerting oxidative chemical reactions in organism. Oxidation is an indispensable bioenergetic process to sustain life. (**b**) Reductive stressors are products of antioxidative enzymes that generate antioxidants in response to regular oxidation activity and increased oxidative stress. (**c**) Antioxidant supplementation attempts to shift the biphasic response from adverse to beneficial phase to maintain nucleophilic tone. This mechanism is called parahormesis.

**Figure 2 biomedicines-08-00406-f002:**
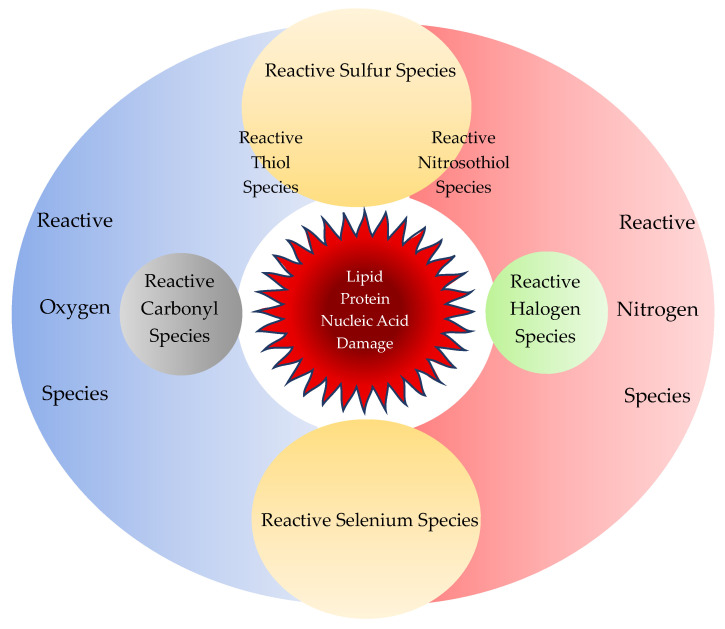
Reactive chemical species. Reactive chemical species comprise of not only reactive oxygen species and reactive nitrogen species, but also reactive sulfur, carbonyl, halogen, and selenium species. Sulfur reacts with oxygen or nitrogen to form reactive thiol or nitrosothiol species, respectively. All reactive chemical species react in concert during regular cellular activity but may cause oxidative stress to damage cellular components such as proteins, lipids, and nucleic acids.

**Figure 3 biomedicines-08-00406-f003:**
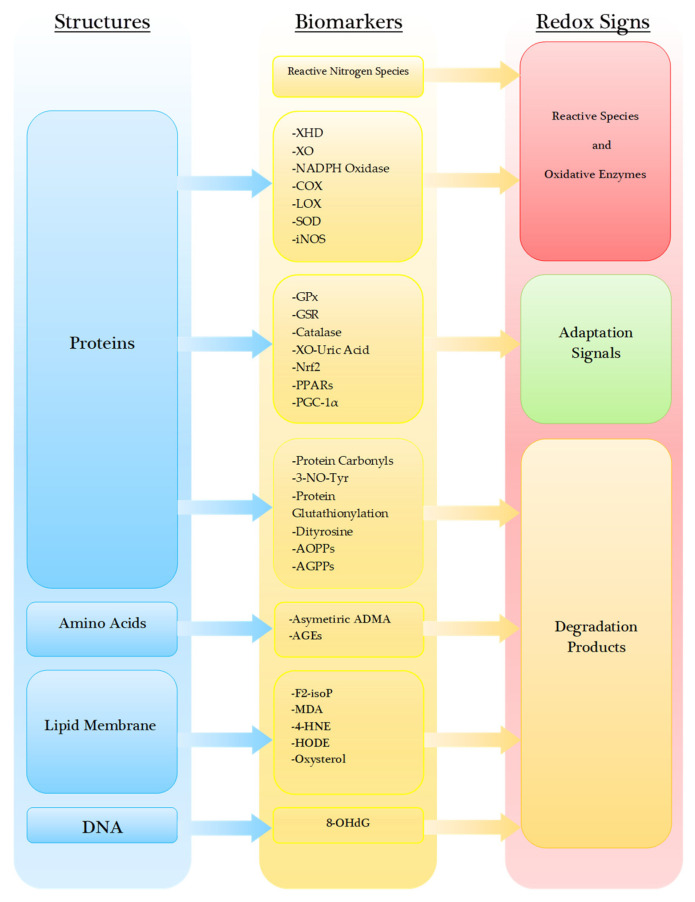
Classification of redox biomarkers according to cellular structures, biomarkers, and their modes of action in redox homeostasis.

**Figure 4 biomedicines-08-00406-f004:**
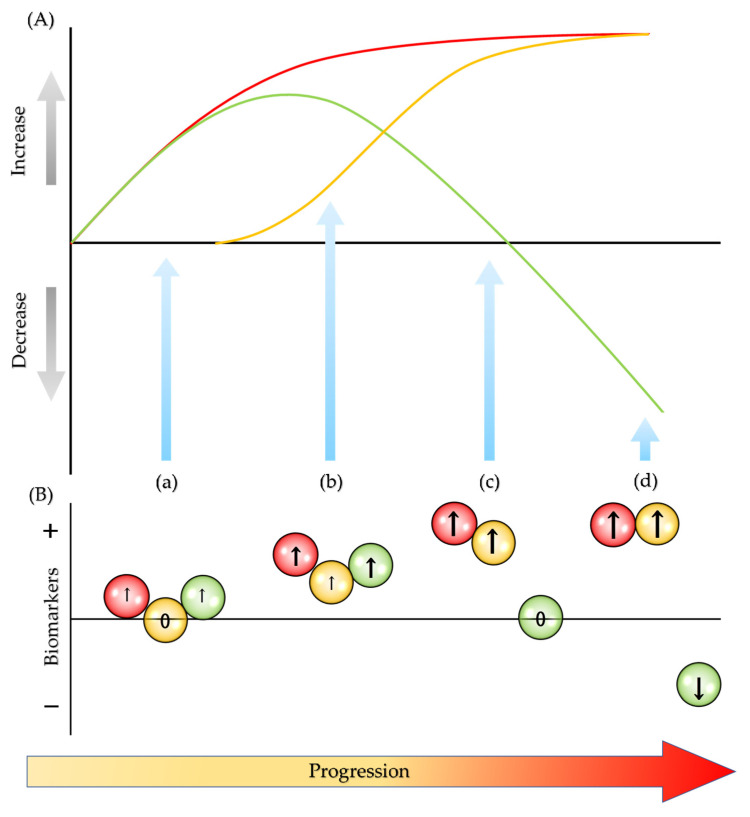
Dynamics of redox components in disease progression. (**A**) **–** (red line): reactive chemical species and activities of oxidative enzymes increase gradually. **–** (orange line): the degradation products increase accordingly upon exhaustion of antioxidative enzymes’ activities. **–** (green line): antioxidative enzymes offset the effects of oxidative enzymes in the early phase, however; the antioxidative activities decline and, finally, fatigue in the later phase, resulting in the exacerbation of inflammation and cellular damage. (**B**) Biomarkers of three redox components may present different values. (a) The levels of oxidative enzyme slightly increase, but the degradation markers stay in a normal range. The antioxidative markers also slightly increase. (b) The oxidative markers further elevate, and the degradation markers start slightly increasing. The antioxidative markers also stay elevated. (c) The oxidative enzyme markers increase, but antioxidative markers return to normal range. The degradation product markers further elevate. (d) The oxidative markers greatly increase, but the antioxidative markers decline. The degradation products markers greatly elevate. 
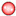
: oxidative biomarkers; 
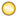
: degradation biomarkers; 
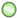
: antioxidative biomarkers; ↑: increase; ↓: decrease; and 0: unchanged.

**Table 1 biomedicines-08-00406-t001:** Licensed disease-modifying drugs in multiple sclerosis [19,20,21,22,23,24,25,26].

Class	Drugs	Indications
Immunomodulators	Interferon beta (IFN-β)	CIS Active RRMS
	Methylprednisolone	Acute flare-ups RRMS Relapse prevention
	Glatiramer acetate (GA)	CIS Active RRMS
	Dimethyl fumarate	Active RRMS
	Diroximel fumarate	CIS RRMS Active SPMS
	Fingolimod	Active RRMS High-active RRMS
	Teriflunomide	Active RRMS
	Siponimod	CIS RRMS SPMS
Immunosuppressors	Alemtuzumab	Active RRMS
	Natalizumab	High-active RRMS
	Ocrelizumab	Active RRMS High-active RRMS PPMS
	Ofatumumab	CIS RRMS Active SPMS
Cytotoxic Agents	Cladribine	High-active RRMS
	Mitoxantrone	High-active RRMS

**Table 2 biomedicines-08-00406-t002:** Reactive chemical species. During regular cellular activity living cells generate numerous reactive chemical species containing oxygen, nitrogen, sulfur, carbonyl, halogen, or selenium. Reactive sulfur species can contain thiols or nitrothiols. Free radicals possess at least one unpaired electron that makes highly reactive and short-lived. Nonradicals are oxidizing chemicals or easily converted to free radicals. Superoxide, reactive sulfur species, or reactive selenium species can be reducing agents as reactive nucleophilic species.

Reactive Chemical Species	
Reactive Oxygen Species (ROS)	Free Radicals
	Superoxide anion (O_2_^−^^•^), Hydroxyl radical (OH^•^), Alkoxyl radical (RO^•^), Peroxyl radical (ROO^•^)
	Nonradicals
	Hydrogen peroxide (H_2_O_2_), Organic hydroperoxide (ROOH), Organic peroxide (ROOR), Singlet oxygen (O_2_^1^Δ_g_), Ozone (O_3_)
Reactive Nitrogen Species (RNS)	Free Radicals
	Nitric oxide radical (NO^•^), Nitrogen dioxide radical (NO_2_^•^)
	Nonradicals
	Nitrite (NO_2_^−^), Nitrate (NO_3_^−^), Nitroxyl anion (NO^−^), Nitrosyl cation (NO^+^), Peroxynitrite (ONOO^−^), Peroxynitrate (O_2_NOO^−^), Nitrosoperoxycarbonate (ONOOCO_2_^−^), Dinitrogen trioxide (N_2_O_3_), Dinitrogen tetraoxide (N_2_O_4_), Nitryl chloride (NClO_2_)
Reactive Sulfur Species (RSN)	Free Radicals
	Thiyl radical (RS^•^), Peroxysulphenyl radical (RSOO^•^)
	Nonradicals
	Hydrogen sulfide (H_2_S), Thiolate anion (RS^−^), Thiol (RSH), Hydropersulfide (RSSH), Disulfide (RSSR), Hydropolysulfide (RSS_n_H), Dialkyl polysulfide (RSSnR), Polysulfide (H_2_Sx), Sulfenate (RSO^−^), Sulfinate (RSO_2_^−^), Sulfonate (RSO_3_^−^), Thiosulmonate (S_2_O_3_^2^^−^), Sulfite (RS_2_O_3_^2−^), Sulfate (SO_4_^2 −^), Thiosulfinate (C_6_H_10_OS_2_), S-nitrosothiols (RSNOs), Nitrosopersulfide (SSNO^−^), Dinitrosylated sulfite adduct (SULFI/NO)
Reactive Carbonyl Species (RCS)	Nonradicals
	Acetaldehyde (CH_3_CHO), Acrolein (Proponel^+^: C_3_H_4_O), Methylglyoxal 4-Hydroxy-nonenal (C_9_H_16_O_2_), 3-Deoxyglucosone (C_6_H_10_O_5_), Glyoxal (C_2_H_2_O_2_), Methylgyoxal (C_3_H_4_O_2_), Electronically excited triplet carbonyls (^3^L=O*)
Reactive Halogen Species (RHS)	Free Radicals
	Atomic chlorine (Cl^•^), Atomic bromine (Br^•^)
	Nonradicals
	Hypochlorite (OCl^−^), Chloramines (RNHCl), Hypobromite (OBr^−^), Hypoiodite (IO^−^), Hypohalogenite (XO^−^; X = F, Cl, Br, or I)
Reactive Selenium Species (RSeS)	Nonradicals
	Selenite (O_3_Se^−2^), Selenate (SeO_4_^2−^), Selenocysteine (C_3_H_7_NO_2_Se), Selenomethionine (C_5_H_11_NO_2_Se)
Reactive Nucleophilic Species	Free Radicals
	Superoxide anion (O_2_^−^^•^)
	Nonradicals
	Hydrogen sulfide (H_2_S), Thiolate (RS^−^), Hydropersulfide (RSS^−^), Disulfide (RSSR), Selenite (O_3_Se^−2^), Selenate (SeO_4_^2−^), Selenocysteine (C_3_H_7_NO_2_Se), Selenomethionine (C_5_H_11_NO_2_Se)

**Table 3 biomedicines-08-00406-t003:** Oxidative stress biomarkers of multiple sclerosis. The redox status can be monitored by the activities of oxidative and antioxidative enzymes and the presence of degradation products derived from cellular components. ↑: increase, ↓: decrease, **-**: unknown.

Classes		Types	Human Samples		Reference
			Blood	CSF	
Reactive Species		Reactive Nitrogen Species	↑	↑	[47]
Oxidative Enzymes		Xanthine Dehydrogenase (XDH)	↓	-	[46]
		Nicotinamide Adenine Dinucleotide Phosphate (NADPH) Oxidase	mixed	-	[48]
		Superoxide Dismutase (SOD)	↑	↓	[49,50,51,52]
		Inducible Nitric Oxide Synthase (*i*NOS)	↑	↑	[18,53,54,55]
		Myeloperoxidase (MPO)	mixed	-	[48]
Antioxidative Enzymes and Transcriptional Factors		Glutathione Peroxidase (GPx)	↑(relapse) ↓(remission)	↓	[56,57,58,59,60,61]
		Glutathione Reductase (GSR)	-	↑	[61,62]
		Catalase	↓	-	[61,62]
		Xanthine oxidase (XO)-Uric Acid	↓	-	[46]
		Nuclear Factor Erythroid 2-Related Factor (Nrf2)	↑	-	[52]
		Peroxisome proliferator-activated receptors (PPARs)	-	↑	[63]
		Peroxisome proliferator-activated receptor gamma coactivator 1-alpha (PGC-1α)	↓	-	[64]
Degradation Products and End Products	Protein	Protein carbonyls	↑	-	[65,66,67,68,69,70,71]
		3-nitrotyrosin (3-NO-Tyr)	↑	-	[68,69,70,72]
		Protein glutathionylation	-	↑	[73]
		Dityrosine	↑	-	[74]
		Advanced oxidation protein products (AOPPs)	↑	-	[49,74,75]
		Advanced glycation end products (AGEs)	↑	↑	[74,75]
	Amino acids	Asymmetric dimethylarginine (ADMA)	↓	-	[76]
	Lipid	F2-isoprostane (F2-isoP)	↑	↑	[77,78,79,80,81]
		Malondialdehyde (MDA)	mixed	↑	[61,82,83,84,85]
		4-hydroxynonenal (4-HNE)	-	↑	[86]
		Hydroxyoctadecadienoic acid (HODE)	↑	↑	[87]
		Oxysterol	mixed	↑	[88]
	DNA	8-dihydro-2′deoxyguanosine (8-oxodG)	↑	-	[56]

**Table 4 biomedicines-08-00406-t004:** Possible redox biomarkers in multiple sclerosis. Reactive chemical species, oxidative enzymes, antioxidants, antioxidative enzymes, degradation products, and end products are potential biomarkers for multiple sclerosis (MS). Diagnostic biomarkers allow early detection and secondary prevention; prognostic biomarkers suggest the likely clinical course; predictive biomarkers predict the response of MS patients to a specific therapy; and therapeutic biomarkers indicate a target for therapy. CIS: clinically isolated syndrome, PPMS: primary progressive MS; RRMS: relapsing-remitting MS, SPMS: secondary progressive MS, mixedMM: mixed population of MS.

Class	Components	Biomarkers			
		Diagnostic	Prognostic	Predictive	Therapeutic
Reactive Chemical Species	Total nitrite (NO_2_^−^)/nitrite (NO_3_^−^) value (tNOx)	PPMS, RRMS, Relapse, SPMS	RRMS	-	-
	*S*-nitrosothiol	RRMS, SPMS	Spinal injury	-	-
Oxidative Enzymes	Superoxide dismutase (SOD)	CIS, RRMS	CIS, RRMS	RRMS	RRMS
	Myeloperoxidase (MPO)	RRMS	RRMS	-	-
	Inducible nitric oxide synthase (*i*NOS)	RRMS	-	-	-
Antioxidants and Antioxidative Enzymes	Xanthine oxidase (XO)-Uric acid	PPMM, RRMM, SPMM	-	-	-
	Selenium	RRMS	-	-	-
	Glutathione reductase (GSR)	mixedMM	MixedMM	-	-
	Catalase	CIS, RRMS	RRMS	-	-
	Thioredoxin-Peroxiredoxin (TRX-PRDX)	MS	-	-	-
	Nuclear factor erythroid 2-related factor (Nrf2)	RRMS	-	RRMS	-
	Peroxisome proliferator-activated receptors (PPARs)	RRMS	-	-	-
	Peroxisome proliferator-activated receptor gamma coactivator 1-alpha (PGC-1α)	PPMS, SPMS	-	-	-
Degradation Products and End Products	Protein carbonyls	RRMS, SPMS	RRMS, SPMS	RRMS	-
	3-nitrotyrosine (3-NO-Tyr)	RRMS, SPMS	-	RRMS	-
	Glutathionylation	Acute attack	-	-	-
	Dityrosine	RRSM	-	-	-
	Advanced oxidation protein products (AOPPs)	CIS, RRMS	RRMS	RRMS	-
	Advanced glycation end products (AGEs)	RRMS	-	-	-
	Asymmetric dimethylarginine (ADMA)	RRMS, SPMS	-	-	-
	F2-isoprostane (F2-isoP)	RRMS, SPMS	SPMS	-	-
	Malondialdehyde (MDA)	RRMS	RRMS	RRMS	-
	4-hydroxynonenal (4-HNE)	PPMS, RRMS, SPMS	-	-	-
	Hydroxyoctadecadienoic acid (HODE)	CIS, RRMS	-	-	-
	Oxocholesterols	MixedMS	SPMS	-	-
	Oxidized low-density lipoprotein (oxLDL)	RRMS, SPMS	-	-	-
	8-OH2dG	RRMS	-	-	-

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
