# Peer review of "Monitoring the Redox Status in Multiple Sclerosis"

_biomedicines, 2020, doi:10.3390/biomedicines8100406_

Round 1

Reviewer 1 Report

This review article provided the information regarding altered components of the reduction-oxidation (redox) homeostasis in four subtypes of patients with multiple sclerosis (MS) and different redox components (e.g., oxidative enzymes, antioxidative enzymes, and degradation products) during the disease progression which could help evaluate the redox status of MS patients and expedite building the most proper personalized treatment plans for such patients. Overall, the manuscript is generally well written. I only have three concerns regarding the levels of detailed description of its methods and context.

  1. Please describe the inclusion/exclusion criteria for retrieving relevant references, such as the methodological quality of the selected studies.
  2. It would be helpful to provide a flow diagram of the selection process of the previous studies.
  3. In the current review paper, some previous findings were stated separately (e.g., see Lines 208-220). Please compare and dicsuss them.

Author Response

Dear Reviewer 1,

Thank you for your reviewing our manuscript with valuable notes, comments, and suggestions. Revision was made as follows:

This review article provided the information regarding altered components of the reduction-oxidation (redox) homeostasis in four subtypes of patients with multiple sclerosis (MS) and different redox components (e.g., oxidative enzymes, antioxidative enzymes, and degradation products) during the disease progression which could help evaluate the redox status of MS patients and expedite building the most proper personalized treatment plans for such patients. Overall, the manuscript is generally well written. I only have three concerns regarding the levels of detailed description of its methods and context.

  1. Please describe the inclusion/exclusion criteria for retrieving relevant references, such as the methodological quality of the selected studies.

As Reviewer 1 suggested, we intended to present a comprehensive review regarding redox biomarkers in multiple sclerosis. During the literature search we noticed that there is a relatively fewer number of research articles with numerous redox biomarkers to cover. Thus, a format of expert review best suit to our purpose rather than systematic review or meta-analysis. Selection criteria were described in Introduction and risk of bias assessment and the criteria were described in Appendix A to increase the evidence level, as suggested by Reviewer 1.

  1. It would be helpful to provide a flow diagram of the selection process of the previous studies.

We made our best effort to cover and include relevant studies according to search terms. Preferred reporting items for systematic reviews and meta-analyses (PRISMA) was not included, as this manuscript is an expert review. We strongly hope that more studies will be available for us to conduct systematic review and meta-analysis in the near future.

  1. In the current review paper, some previous findings were stated separately (e.g., see Lines 208-220). Please compare and dicsuss them.

The paragraph was revised to present significance of the findings and their possible applicability briefly.

We all appreciate the Reviewer’s kind help and expert opinions indispensable to improve the quality of our manuscript.

Best regards,

Masaru Tanaka, M.D., Ph.D.

Reviewer 2 Report

Multiple sclerosis (MS) is an autoimmune disease, associated with central nervous system (CNS) inflammation, demyelination, and axonal loss. Myelin, a multilayer membranous that covers nerve fibers, is essential for rapid impulse conduction. Oligodendrocytes that are generated either from CNS-resident oligodendrocyte progenitor cells (OPCs) or subventricular zone-derived neural stem cells (NSCs) are the myelinating cells of the CNS. The adult CNS maintains a certain endogenous potential to repair myelin damage. However, this process often fails as MS progresses. The origin of this failure is not fully understood, but it is likely to relate to progenitors/stem cells' arrestment in a quiescent state, incapable of generating new oligodendrocyte. Current treatments for MS are immunomodulatory or immunosuppressive medications, with little to no effect on myelin restoration. Recent studies have provided proof-of-principle that CNS remyelination can be promoted either via enhancing endogenous remyelination or by transplanting myelinating cells. Curcumin, a natural polyphenolic compound, has been shown to have therapeutic properties in several neurodegenerative diseases.  However, increasing efforts to employ current anti-inflammatory agents in MS clinical trials are emerging to produce significant clinical success. Yet, there is a need to explore anti-inflammatory natural products, which target neuroinflammatory pathways relevant to MS pathogenesis. Numerous evidences suggest that plant polyphenols may have therapeutic benefits in regulating oxidative stress and providing neuroprotection in many neurodegenerative diseases, including multiple sclerosis (MS). However, these mechanisms are not yet completely understood. In this study, we investigated the effect of olive leaf polyphenols on oxidative stress through oxidation marker level and activity (TBARS, SOD, and GPX) and their protein expression (SOD1, SOD2, and GPX1), as well as the protein expression of Sirtuin 1 (SIRT1) and microglia markers (Iba-1, CD206, and iNOS) and myelin integrity (proteolipid protein expression) in the brain of rats with induced experimental autoimmune encephalomyelitis (EAE) and subjected to olive leaf therapy. Consistent to this notion, polyphenols are strong antioxidants with characteristics that are of beneficial therapeutic values for their development as candidates targeting neurodegenerative and oxidant-induced diseases. Thus, interplay and  redox coordination of polyphenols is an emerging area of reserach interest in anticancer and antidegenerative therapeutics. Moreover, hormetic responses occur in a wide range of biological models for a large and diverse array of endpoints. Particular attention has been given to providing an assessment of the quantitative features of the dose-response relationships and underlying mechanisms that could account for the biphasic nature of the hormetic response after exposure to metals. The hormetic dose response should be seen as a reliable feature of the dose response in biology and medicine and appears to have an important impact on the estimated effects of a variety of chemical and biological agents on brain pathophysiology and stress resistance mechanisms to oxidative and inflammatory insult and damage.

This is an interesting paper.  The study is well-conceived and well-executed. This reviewer is satisfied with the significance of this study, the care in which the study was performed, and the implications of the results for human health.  However, although the results presented are convincing, the work raises some concerns which will need to be addressed. The questions posed are of extremely high interest, but the paper does not give adequate definitive information, therefore pending addressing some minor question is possible to accept for publication.

Minor mandatory concerns:

  1. Preconditioning signal leading to cellular protection through Hormesis is an important redox dependent aging-associated neurodegenerative/ neuroprotective issue. This aspect should be highlighted in the discussion and references properly added (See Brunetti et al., 2020 Int J Mol Sci. Apr 8;21(7):2588; Di Rosa et al., 2020 Int J Mol Sci. May 29;21(11):3893; Scuto et al., 2019 Int J Mol Sci. Dec 31;21(1):284;

  1. Given the relationship between polyphenol compounds, redox status and the vitagene network and its possible biological relevance in neuroprotection, Authors while interpetrating results should discuss appropriately this aspect and make proper connection with emerging principles of hormesis ( Calabrese et al., 2010, Antiox. Redox Signal 13,1763; Calabrese et al., Ageing Res Rev. 2018, Mar;42:40-55); Leri et al., Int J Mol Sci. 2020, 21(4):1250; 178; Siracusa et al., Antioxidants 2020, 9(9):E824).

Author Response

 Dear Reviewer 2,

Thank you for your reviewing our manuscript with valuable notes, comments, and suggestions. Revision was made as follows: 

Minor mandatory concerns:

  1. Preconditioning signal leading to cellular protection through Hormesis is an important redox dependent aging-associated neurodegenerative/ neuroprotective issue. This aspect should be highlighted in the discussion and references properly added (See Brunetti et al., 2020 Int J Mol Sci. Apr 8;21(7):2588; Di Rosa et al., 2020 Int J Mol Sci. May 29;21(11):3893; Scuto et al., 2019 Int J Mol Sci. Dec 31;21(1):284;

We appreciate your valuable suggestions. We added a brief description of hormesis and parahormesis in the section of Antioxidants including suggested references.

  1. Given the relationship between polyphenol compounds, redox status and the vitagene network and its possible biological relevance in neuroprotection, Authors while interpetrating results should discuss appropriately this aspect and make proper connection with emerging principles of hormesis ( Calabrese et al., 2010, Antiox. Redox Signal 13,1763; Calabrese et al., Ageing Res Rev. 2018, Mar;42:40-55); Leri et al., Int J Mol Sci. 2020, 21(4):1250; 178; Siracusa et al., Antioxidants 2020, 9(9):E824).

Thank you for your valuable comments. We added a description of benefit of the Mediterranean diet including polyphenols with suggested references. We were not able to delve into the exciting areas of research, but really hope that this manuscript succeeds in presenting the introductory concept as a part of redox biomarkers in multiple sclerosis.

We all appreciate the Reviewer’s kind help and expert opinions indispensable to improve the quality of our manuscript.

Best regards,

Masaru Tanaka, M.D., Ph.D.

Round 2

Reviewer 2 Report

The paper can be accepted. Authors effort has been much appreciated